# *Bacillus thuringiensis* Cry1A Insecticidal Toxins and Their Digests Do Not Stimulate Histamine Release from Cultured Rat Mast Cells

**DOI:** 10.3390/biology14010015

**Published:** 2024-12-27

**Authors:** Hisashi Ohto, Mayumi Ohno, Miho Suganuma-Katagiri, Takashi Hara, Yoko Egawa, Kazuya Tomimoto, Kosuke Haginoya, Hidetaka Hori, Yuzuri Iwamoto, Tohru Hayakawa

**Affiliations:** Graduate School of Science and Technology, Niigata University, 8050 Ikarashi 2 No-Cho, Nishi-ku, Niigata 950-2181, Japan

**Keywords:** *Bacillus thuringiensis*, insecticidal Cry1A toxins, simulated digestive fluids, rat mast cells, RBL-2H3, safety of genetically modified crops

## Abstract

The soil bacterium *Bacillus thuringiensis* (Bt) produces protein crystals during sporulation. Cry toxins, a major component of these protein crystals, are insecticidal proteins with pore-forming activity. Cry toxins are environmentally friendly biopesticides used to create Bt-incorporating genetically modified crops (BT-GMCs) that confer resistance to insect pests. However, public acceptance of BT-GMCs is lacking. A major concern over BT-GMCs is the allergenicity of Cry toxins. Specifically, the pore-forming activity of Cry toxins may directly induce intestinal mast cells to release histamine. In this study, three lepidopteran-specific Cry toxins, Cry1Aa, Cy1Ab, and Cry1Ac, were treated with simulated digestive fluids under different conditions. Cry1A toxins exhibited differing sensitivity to simulated digestive fluids, and digests of Cry1A toxins contained the α4–α5 helical hairpin of domain I, which is thought to be the transmembrane domain. To determine whether Cry1A toxins directly induce histamine release, intact and digested Cry1A toxins were applied to RBL-2H3 cultured rat mast cells. However, no histamine secretion from these mast cells was detected, even when samples were analyzed using HPLC. The results of our study provide important data supporting the safety of Cry1A toxins and potentially BT-GMCs, and reducing anxiety over their safety should help increase public acceptance.

## 1. Introduction

*Bacillus thuringiensis* (Bt) is a Gram-positive soil bacterium that produces protein crystals during sporulation. Cry toxins, a major component of these protein crystals, are insecticidal proteins that share a similar three-domain architecture (domains I, II, and III). Cry toxins are generally considered to be α-pore-forming toxins in which α-helical hairpins of domain I form pores in target cell membranes. Cry toxins have a narrow insecticidal spectrum, and formulations containing these toxins are used worldwide as environmentally friendly biopesticides [1]. However, Cry toxins are inactivated by sunlight and removed from foliage relatively quickly after application. To overcome this disadvantage, Cry toxins have been incorporated into genetically modified crops (BT-GMCs) to confer resistance to insect pests. The area covered by fields planted with GMCs is growing significantly and now cumulatively covers more than 4 billion acres worldwide. BT-GMCs now constitute 25% of the various GMCs, and 51 nations allow the culture of BT-GMCs [2,3]. The area planted with BT-GMCs has increased significantly, especially in developing countries, and covered 190 million ha in 26 countries in 2018. BT-GMCs currently account for 30, 78, 29, and 76% of maize, soybean, rapeseed, and cotton varieties, respectively [2]. Importantly, since their introduction in 1996, there have been no documented adverse effects associated with BT-GMCs in humans or animals. The lack of adverse effects is due to rigorous safety evaluations conducted prior to commercialization. This safety assessment process is designed to determine the allergenic potential of a crop and prevent the introduction of known or novel allergens into the food supply. Such safety assessments are further strengthened by ensuring that the levels of endogenous allergens in a crop are not significantly altered compared with the non-BT-GMC counterpart. This is not required for non-GMCs but provides assurance to the public that foods obtained from BT-GMCs are safe for consumption. The possibility that BT-GMCs could be associated with increased risk of food allergy was acknowledged in 1992 by the US Food and Drug Administration. However, to date, no evidence has emerged indicating that BT-GMCs have caused food allergies or pose any risks to consumers.

In contrast to the formulation sold worldwide, public acceptance of BT-GMCs is lacking. In 2000, it was reported that patients who ingested a BT-GMC corn product known as Star-Link (containing the *cry9Ca* gene, which encodes a 60-kDa Cry toxin that is active against both Lepidoptera and Coleoptera) developed an allergic reaction [4]. This so-called “Star-Link incident” diminished the otherwise growing public acceptance of BT-GMCs [5,6]. The US Environmental Protection Agency reported that Cry9Ca has the potential to induce allergy because the protein is stable at high temperatures and under acidic conditions and resistant to digestive enzymes [4]. Later research showed that the affected patients did not have IgE specific to Cry9Ca but were instead allergic to maize proteins [7]. The allergenicity of BT-GMCs has also been disputed in several reports. For example, Reiner et al. [8] found that GMC-BT maize had no adjuvant effect on allergic responses in mice. Cao et al. [9] found that Cry1C protein from GMC-BT rice was not allergenic, and a bioinformatics analysis conducted by Randhawa et al. [10] found no amino acid sequences predictive of allergenicity. In addition, many reports have supported the safety of *B. thuringiensis* Cry toxins in mammals, especially after the commercialization of BT-GMCs. For example, Nishiitsutsuji-Uwo et al. [11] reported that cultured NT-368 insect cells appeared swollen with destroyed mitochondria after a 15 min incubation with Cry toxin, but S-180 cultured mouse cells exhibited no adverse effects during a 4 h incubation. Thomas and Ellar [12] demonstrated insecticidal activity of Cry toxin against *Pieris brassicae* larvae at 2 μg, but they found no signs of toxicity in mice treated by venous administration of 500 μg of toxin. Chowdhury et al. [13] fed pigs for 4 weeks with the GMC-BT maize product Bt11, which contains Cry1Ab. They detected undigested Cry1Ab in the stomach, duodenum, appendix, ileum, and rectum, but the pigs gained weight normally. On the other hand, there have been recent reports of the allergenic potential of Cry toxins. Santos-Vigil et al. [14] reported that Cry1Ac protein administered via the intra-gastric route was capable of inducing moderate allergy-related immune responses in mice. As Cry toxins used in insecticide formulations and BT-GMCs are important tools in integrated pest management practices to reduce the use of chemical insecticides, accumulation of safety data for Cry toxins is needed.

In this study, we digested several lepidopteran-specific Cry toxins (Cry1Aa, Cry1Ab, and Cry1Ac) commonly used in BT-GMCs using simulated digestive fluids. The resulting digests were analyzed by sodium dodecyl sulfate–polyacrylamide gel electrophoresis (SDS-PAGE) and Western blotting to determine which α-helical and β-sheet structures of the toxin remained in the digests. The Cry toxins and corresponding digests were then tested for the potential to induce histamine release from rat mast cells. Mast cells are a critical component of the immune system, known for their role in allergic reactions and the release of histamine in response to environmental triggers, making them an ideal model for this study.

## 2. Materials and Methods

### 2.1. Cry1A Toxins and Cultured Mast Cells

Three lepidopteran-specific Cry toxins commonly used to generate various BT-GMCs, Cry1Aa, Cry1Ab, and Cry1Ac, were used in this study. To avoid cross-contamination, each toxin protein was produced separately using *Bacillus thuringiensis* (Bt) strains that produce only one type of toxin. Bt *sotto* T84A1, which produces Cry1Aa, and Bt *kurstaki* HD-73, which produces Cry1Ac, were cultured as previously described [15]. Bt sotto T84A1 was kindly provided by Dr. M. Ohba, Kyushu University, Japan. In the case of Cry1Ab, we used a recombinant *Escherichia coli* with plasmid vector *pYD4.0* containing the *Cry1Ab* gene [16], kindly provided by Dr. K. Kanda, Saga University, Japan. All Cry1A toxins were harvested as protein crystals or inclusions and solubilized in alkaline buffer (50 mM Na_2_CO_3_, 10 mM dithiothreitol). After removal of insoluble material by centrifugation, the alkali-soluble toxins were trypsin activated and then purified by DEAE-Sepharose column chromatography, as previously described [17,18]. Protein concentration was estimated using Bio-Rad protein assay dye reagent (Bio-Rad Laboratories, Hercules, CA, USA) with bovine serum albumin as the standard. Proteins were analyzed by SDS-PAGE as previously described [19], and Tricine SDS-PAGE was performed as previously described [20]. Protein bands were visualized with Coomassie brilliant blue (CBB) staining, and molecular size was determined using Quantity One analytical software (ver. 16.0, Bio-Rad).

RBL-2H3 rat mast cells were maintained in RPMI-1640 medium (Sigma, St. Louis, MO, USA) supplemented with 10% (*v*/*v*) of fetal bovine serum and L-glutamine (Sigma, R5886), as previously described [21]. Cells were incubated in a 5% CO_2_ incubator at 37 °C and passaged every 3 days.

### 2.2. Digestion of Cry1A Toxins Using Simulated Digestive Fluids

Cry1A toxins were digested using simulated gastric fluid (SGF) and then simulated intestinal fluid (SIF). SGF consisted of 0.32% (*w*/*v*) pepsin, 0.2% (*w*/*v*) NaCl, and 0.7% (*v*/*v*) HCl, and the pH was adjusted as indicated (between pH 2 and 6) using 1 M citrate or glycine buffer. In general, the pH of human gastric fluid varies from pH 2 to 6 in a time-dependent manner after food ingestion [22]. Briefly, 350 μL of toxin solution was mixed with 150 μL of SGF solution and incubated at 37 °C for the indicated time. The reaction was stopped by immersing the test tubes in liquid N_2_, and the digests were then analyzed by SDS-PAGE. Bovine serum albumin (BSA) was used as a control. For Cry1Aa, the digests were further analyzed by Western blotting, as described in Section 2.3 below.

Cry1A toxins digested with SGF were further digested (at 37 °C for 4 h) with SIF containing 1.0% (*w*/*v*) pancreatin in 100 mM NaHCO_3_ (pH 8.0). Namely, 500 μL of SGF digestion solution as described above was mixed with 500 μL of SIF and adjusted to pH 8.0 with 1 M NaOH. In the sequential digestion with SGF and SIF, we only used SGF at pH 2 or 4, as higher values (e.g., pH 6) are considered abnormal in gastric physiology. The resulting Cry1A toxin digests were then analyzed by Western blotting using specific antiserum as described in Section 2.3 below.

### 2.3. Western Blotting

Cry1A toxins and corresponding digests separated by SDS-PAGE were electroblotted onto PVDF membranes as previously described [18]. To characterize the structures of the digested toxins, specific rabbit antisera were raised against the Cry1Aa polypeptides α2–α3, α4–α5, and α6–α7 helices from domain I, polypeptides β1-β5, and the β6-β11 sheets from domain II and domain III [23]. Bound antibodies were detected by incubation with peroxidase-conjugated goat anti-rabbit IgG for 1 h at 4 °C, followed by visualization using the ECL Western blotting detection system (GE Healthcare, Buckinghamshire, UK).

### 2.4. Microscopic Observation of Rat Mast Cells Treated with Cry1A Toxins

After purification by DEAE-Sepharose column chromatography, trypsin-activated Cry1A toxins were labeled with the fluorescent dye Atto-465 (ATTO Tec, Siegen, Germany) according to the manufacturer’s instructions. Unreacted dye was removed by dialysis against phosphate-buffered saline. A suspension of RBL-2H3 cells (1 × 10^6^ cells) was incubated with labeled Cry1A toxin at a final concentration of 10 nM (0.6 μg/mL) or 100 nM (6 μg/mL) for 60 min at 37 °C in the dark. Cell-bound fluorescence was observed under an inverted fluorescence microscope (IX-71, Olympus, Tokyo, Japan) equipped with BP470-490 and BA510-550 filters for excitation and emission, respectively, and photographs were taken at ×400 magnification using an Olympus U-Plan APO objective lens (40×) with an Olympus CCD camera. The sensitivity, exposure compensation, and exposure mode were set to ISO200, 0, and auto exposure for fluorescence, respectively. Fluorescence intensity was determined using a fluorescence meter (RF-5300PC, Shimadzu Corp., Kyoto, Japan) and a standard calibration curve. Statistical significance was evaluated using Student’s *t* test. Cells were also observed under visual light, and cell viability was estimated using trypan blue staining. Standard deviations were calculated from triplicate experiments.

### 2.5. Detection of Histamine Release from Cry1A Toxin-Treated Rat Mast Cells

A suspension of RBL-2H3 cells (1 × 10^6^ cells) was seeded in a Petri dish (35 mm Ф, Nalgene Nunc International, Rochester, NY, USA) and incubated in 3 mL of RPMI-1640 medium for 2 days at 37 °C. RBL-2H3 cells on the dish were washed three times with 1 mL of freshly prepared Hanks’s buffer [24] and then incubated in 1 mL of Hanks’s buffer containing the indicated toxin or corresponding digest. The viability of the cells incubated with the toxin was determined using trypan blue staining. The calcium ionophore A23187 (0.5 ng/mL) and BSA (100 μg/mL) were used to estimate histamine secretion in positive and negative controls, respectively.

Histamine released from rat mast cells was extracted and detected as previously described [25]. Histamine was visualized using *o*-phthalaldehyde (OPA), and the resulting fluorescence was measured using a fluorescence meter (RF-5300PC, Shimadzu) at 360 nm for excitation and 440 nm for emission. The amount of histamine released from rat mast cells was expressed as a percentage of the total histamine in rat mast cells. Fluorescence values were adjusted by subtracting the fluorescence of SGF, SIF, Hanks’s buffer, or the Cry1A toxin suspension used. Statistical significance was evaluated using Student’s *t* test. In addition, high-performance liquid chromatography (HPLC) was used to analyze histamine released from rat mast cells. Briefly, histamine samples prepared according to the method of Shore et al. [25] were mixed with the same amount of OPA reagent and incubated for 1 min at room temperature. After the addition of 60 μL of 0.1 M sodium acetate, the mixture was immediately injected onto a Pu2080 HPLC (Hitachi, Tokyo, Japan) equipped with a Gemini C18 (Φ45 × H150 mm, 110A, 5 μm) LC column (Phenomenex, Torrance, CA, USA). Fluorescence of histamine was determined by chemiluminescence of FP-2020 (JASCO, Tokyo, Japan). The sample was loaded with solvent A (tetrahydrofuran/methanol/10 mM phosphate buffer [pH 7.2] = 1:19:180 [*v*/*v*]) for 30 min, followed by 80% (*v*/*v*) methanol, at a flow rate of 1.5 mL/min. The column temperature was maintained at 40 °C, and the sample was degassed prior to injection.

## 3. Results

### 3.1. Digestion of Cry1A Toxins with SGF at Varying pH

Cry1A toxins were digested with SGF at varying pH values for 15 min and analyzed by SDS-PAGE. The pH of SGF was adjusted with 1 M glycine (pH 2–4) or citrate buffer (pH 3–6). As shown in Figure 1, BSA (a control protein) was vigorously digested at pH 2, 3, and 4, but the extent of digestion was relatively low at pH 5 and 6, with 40 and 97% of the BSA remaining undigested, respectively. This result suggested that pepsin in SGF is highly active in the pH range 2–4.

By contrast, the Cry1A toxins exhibited a different digestion pattern to that of BSA. Cry1Aa was most sensitive to digestion by SGF, which left almost no undigested protein remaining at pH 2; by contrast, digestion of Cry1Ab and Cry1Ac at pH 2 left significant amounts of undigested protein (Figure 1). At pH 3, the extent of the Cry1A digestion appeared to be reduced in comparison with the digestion of BSA (Figure 1). Interestingly, a polypeptide of ~39 kDa was detected in all Cry1A toxin digests (Figure 1). In this study, Cry1A toxin digestion was stopped relatively early (15 min), and the profile may be different in samples with prolonged incubation. We used the 15 min digestion sample because it was appropriate to compare the sensitivity to SGF digestion and to efficiently recover a variety of potentially undigested polypeptides.

### 3.2. Western Blotting Analysis of Cry1Aa SGF Digests

Cry1Aa toxins digested with SGF were further characterized by Western blotting with specific antiserum raised against each of six Cry1Aa polypeptides. Interestingly, samples of Cry1Aa digested at pH 2 showed almost no protein by SDS-PAGE/CBB staining, but many clear signals, especially at approximately 39 kDa and 14 kDa, were detected by Western blotting with anti-α2–α3 helix antiserum (Figure 2).

The signal at approximately 39 kDa may represent the same band detected by SDS-PAGE/CBB staining as a 39 kDa polypeptide (Figure 1), and a similar signal was detected using all six specific antisera (Figure 2). The 39 kDa polypeptide was thought to span all three domains and be relatively resistant to SGF digestion. The signal at 14 kDa was also detected using the anti-α4–α5 helical hairpin antiserum, suggesting that the polypeptide extends over the α2–α5 helices of domain I. In addition, polypeptides ranging from 17 to 23 kDa were detected with both the anti-α2–α3 and anti-α4–α5 antisera but not with the other antisera (Figure 2). These observations suggest that the α2–α5 region in domain I is more resistant to SGF digestion than the other parts of Cry1Aa.

### 3.3. Sequential Digestion of Cry1A Toxins Using SGF Followed by SIF

Cry1A toxins were digested with SGF at pH 2 or 4 for 1, 15, or 60 min, followed by SIF digestion (pH 8) for 4 h. After separation by SDS-PAGE, the digests were transferred onto PVDF membranes and analyzed by Western blotting using anti-α4–α5 antiserum [23]. As mentioned in Section 3.1, Cry1Aa was the most sensitive to SGF, and digestion at pH 2 for 15 min generated only a 14 kDa polypeptide (Figure 3). By contrast, Cry1Ab and Cry1Ac appeared to be resistant to SGF digestion at pH 2, which generated many polypeptides, including the 14 kDa polypeptide, as well as undigested intact toxin even after 60 min of incubation (Figure 3). The polypeptides generated by SGF digestion were further digested by subsequent treatment with SIF for 4 h, but some polypeptides remained undigested, especially for Cry1Ab and Cry1Ac (Figure 3). SGF digestion at pH 4 was poor, as evidenced by a large amount of intact toxin remaining for all Cry1A toxins, even after subsequent digestion with SIF (Figure 3). Since activated Cry1A toxins are generated by trypsin treatment, intact toxins and presumably digested toxin fragments may be relatively resistant to SIF digestion.

In this study, we demonstrated that the Cry1A toxins examined have differing sensitivities to simulated digestive fluids. At least some of the Cry1A polypeptides remaining after SGF and/or SIF digestion included α4–α5 of domain I. According to the umbrella model, the α4–α5 helical hairpin of domain I is the transmembrane domain of the pore formed by Cry1A toxins [26]. We considered the possibility that these Cry1A polypeptides might interact with a variety of cells and therefore examined their effect on mast cells.

### 3.4. Microscopic Observation of RBL-2H3 Cultured Rat Mast Cells Treated with Cry1A Toxins

RBL-2H3 cells were incubated with 10 nM or 100 nM Atto-645-labeled Cry1A toxins for 60 min in the dark. Subsequent microscopic observation revealed that the treatment did not induce any significant morphological changes, such as swelling or bursting of the cells (Figure 4A). Furthermore, the amount of each Atto-645-labeled Cry1A toxin bound to the cells was similar to that of Atto-645-labeled BSA as a control (Figure 4A). In the 10 nM treatment, the amount of Cry1Aa, Cry1Ab, Cry1Ac, and BSA bound to the cells was estimated as 0.15, 0.22, 0.38, and 0.60 pmol, respectively (Figure 4B). An experiment using a higher protein concentration (100 nM) showed a similar trend, with estimated bound amounts of Cry1Aa, Cry1Ab, Cry1Ac, and BSA of 3.0, 2.6, 4.0, and 5.5 pmol, respectively (Figure 4B). Thus, the binding of both labeled Cry1A toxins and labeled BSA to RBL-2H3 rat mast cells was considered non-specific, as the cells did not appear to be saturated even at 100 nM treatment. In a series of microscopic observations, cell viability was 97.1 ± 1.6% (n = 3), as determined by trypan blue staining. The binding of Cry1A polypeptides generated by SGF and/or SIF digestion to RBL-2H3 rat mast cells was not evaluated in this study. It is of great interest to investigate the interaction between specific toxin polypeptides that may be generated by digestion and mast cells in the future.

### 3.5. Potency of Cry1A Toxins and Corresponding Digests in Inducing Histamine Release from Rat Mast Cells

RBL-2H3 cells were incubated with the respective Cry1A toxins, and histamine released from the cells was quantified using the OPA method. During these experiments, we often observed significant aggregation of toxin and digested toxin molecules, particularly at high concentrations of Cry1Ab and Cry1Ac. We therefore decided to avoid the experiment if such aggregation was observed. RBL-2H3 cells appeared to release a small amount of histamine following treatment with each of the undigested Cry1A toxins (Figure 5A). The amount of histamine released was estimated at approximately 3% of the total histamine present. This was similar to the amount of histamine released by cells treated with BSA (2%), indicating that the level was close to background. By contrast, treatment with the ionophore A23187 at 0.5 ng/mL as a positive control induced significant histamine release from the cells, with the amount estimated at approximately 30% of the total (Figure 5A). Similarly, when RBL-2H3 mast cells were mixed with Cry1A polypeptides generated by SGF digestion (pH 2 and 4) instead of undigested Cry1A toxins, almost no histamine was released (Figure 5B). Similarly, no histamine release was detected from cells treated with Cry1A polypeptides generated by sequential digestion with SGF followed by SIF (Figure 5C). These results suggested that neither intact Cry1A toxins nor Cry1A digests induce rat mast cells to release histamine.

Incidentally, the OPA method is sensitive not only to the amine moiety of histamine but also to the amino base of the amino acids. Therefore, histamine released from rat mast cells was further analyzed by HPLC. Authentic histamine (0.5 μmol) showed a symmetrical single peak with an intensity of 210 mV and retention time (RT) of approximately 19.6 min, but no similar peak was detected in Hanks’s buffer (Figure 6A,B). Mast cells sonicated in Hanks’s buffer were then analyzed, and a histamine peak was detected with an overlapping methionine peak (Figure 6C). Taking into account that these two peaks interfered with each other, the net intensity of the histamine peak was estimated as 20 to 30 mV. Analysis of the culture supernatant of mast cells treated with the ionophore A23187 revealed a single symmetrical peak of approximately 25 mV at an RT of 19.6 min (Figure 6D). This suggests that most of the histamine was released from the mast cells in response to A23187 treatment. An extract of unstimulated mast cells showed the histamine peak as a shoulder of the methionine peak (Figure 6E). The height of the histamine peak was estimated at 5 mV after subtraction of the interference from the methionine peak. By contrast, the HPLC chromatogram of an extract of mast cells treated with the Cry1Aa SGF digest also showed a histamine peak very similar to that of unstimulated cells (Figure 6F). We thus concluded that almost no histamine is released by the mast cells following treatment with Cry1Aa SGF digest.

## 4. Discussion

Since the “Star-Link incident” in 2000, there has been a growing interest in the allergenicity of Cry toxins. In food allergy, various undigested polypeptides that are resistant to protein digestion generally bind to B cells that recognize the polypeptides as antigens, and the B cells then produce IgE specific to the polypeptide. This IgE then binds to the IgE receptor, Fcε, located on the surface of histamine-producing mast cells [27,28]. Subsequent binding of the antigen, such as proteinase-resistant peptide, to the two Fcε receptors forms a bridge-like structure that activates the mast cells, which then secrete inflammatory mediators such as histamine and leukotrienes that trigger an allergic reaction. However, in patients affected during the Star-Link incident, no Cry9Ca-specific IgE, which generally would play an important role in allergic reactions, was detected [7]. This suggests that either the patients were not allergic to Cry9Ca or that Cry9Ca does not sensitize the human immune system. In the latter case, a patient’s mast cells should respond to Cry9Ca independent of the presence of IgE. This is not surprising, as some exceptions show that histamine release from mast cells can be induced by a variety of stimuli that do not involve interaction with IgE [29,30,31]. In addition, Guerrero et al. [32] suggested that Cry1A toxin directly stimulates mouse lymph cells to release cytokines. Thus, whether the Cry toxins and/or their corresponding digests generated by digestive fluid can directly induce histamine release from intestinal mast cells remains controversial. Cry toxins are generally classified as α-pore-forming toxins in which the α-helical hairpins of domain I form pores in the target cell membrane, and pore formation by many Cry toxins has been reported in artificial lipid bilayers lacking specific receptors [33,34,35,36,37,38,39,40]. Several studies have also reported that Cry toxin domain I polypeptides with an α-helical structure are often resistant to proteolytic digestion, exhibit pore-forming activity in plasma membranes, and can be allergenic [41,42].

In the present study, we first examined the digestion of three Cry1A toxins using simulated digestive fluids. Digestion with SGF revealed that compared with BSA, all three Cry1A toxins were relatively resistant to pepsin (Figure 1). Among the three Cry1A toxins, Cry1Aa was relatively sensitive, and most of the toxin was degraded at pH 2. However, a significant amount of undigested toxin was observed following digestion of Cry1Ab and Cry1Ac (Figure 1). Western blotting analysis using antisera raised specifically against six different parts of Cry1Aa [23] revealed that domain I, particularly the region spanning the α2–α5 helices, is the most resistant to SGF digestion (Figure 2). In addition, sequential digestion using SGF followed by SIF revealed that the undigested polypeptides of all three Cry toxins examined contained the α4–α5 helical hairpin of domain I (Figure 3). Because domain I, specifically the α4–α5 helical hairpin, is the transmembrane domain of Cry1A toxin pores, the Cry1A digests may retain the ability to form pores and/or stimulate mast cells to release histamine.

Notably, no morphological changes such as swelling or bursting were observed in RBL-2H3 rat mast cells treated with Atto-645-labeled Cry1A toxins (Figure 4). Indeed, the Atto-645-labeled Cry1A toxins bound to the cells, but the binding appeared to be non-specific, as the cells remained unsaturated even following treatment with toxin at 100 nM (Figure 4). Furthermore, neither undigested Cry1A toxins nor Cry1A digests prepared using different digestive fluids induced significant histamine release from rat mast cells (Figure 5 and Figure 6). We therefore concluded that Cry1Aa, Cry1Ab, and Cry1Ac toxins and corresponding digests generated using simulated digestive fluids do not induce measurable histamine release in rat mast cells lacking specific IgE. On the other hand, because high concentrations of Cry1Ab and Cry1Ac that aggregate during toxin preparation were not tested in this study (Figure 5), it remains possible that such higher concentrations of toxins, particularly Cry1Ab and Cry1Ac, may induce histamine release from mast cells. It would be interesting in the future to evaluate whether toxin aggregation has an effect on mast cells.

## 5. Conclusions

In this study, the allergenicity of Cry toxins (Cry1Aa, Cry1Ab, and Cry1Ac) commonly used in BT-GMCs was evaluated. Cry1A toxins were relatively resistant to simulated digestive fluids compared to BSA as a control, and in particular the transmembrane α4–α5 of domain I, which may retain the ability to form pores, was most resistant to digestion. This suggests that Cry1A toxins and corresponding digests form pores in the cell membrane of rat mast cells, at least at a low frequency. On the other hand, no specific binding of Cry1A toxins to cultured rat mast cells was detected by fluorescence microscopy. Furthermore, no significant histamine release was detected from mast cells treated with Cry1A toxins and these digests by the OPA method and even HPLC analysis. This suggests that pore formation by the toxin in the cell membrane of rat mast cells does not stimulate histamine release or that pore formation by these polypeptides is limited or does not occur. Our results provide useful data for evaluating the non-allergenicity of Cry1A toxins.

## Figures and Tables

**Figure 1 biology-14-00015-f001:**
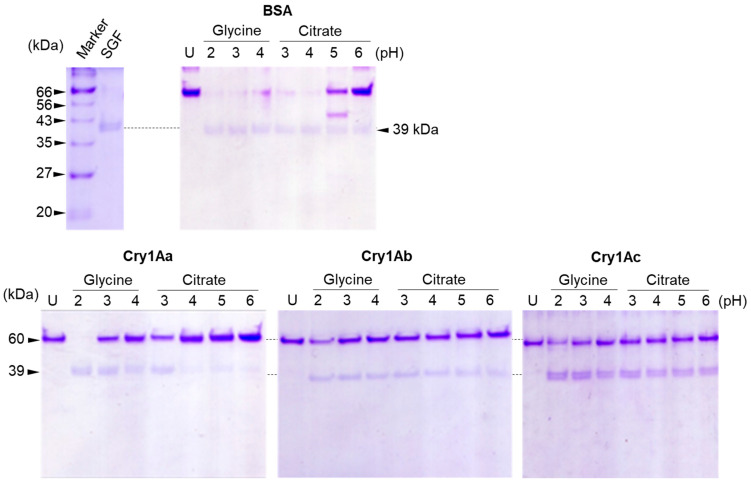
Digestion of Cry1A toxins using SGF at varying pH values. A total of 100 μg of each Cry1A toxin was digested with SGF in glycine buffer at pH 2, 3, and 4 or citrate buffer at pH 3, 4, 5, and 6. The resulting digests were separated by 14% SDS-PAGE and visualized by CBB staining. U: undigested Cry1A toxin.

**Figure 2 biology-14-00015-f002:**
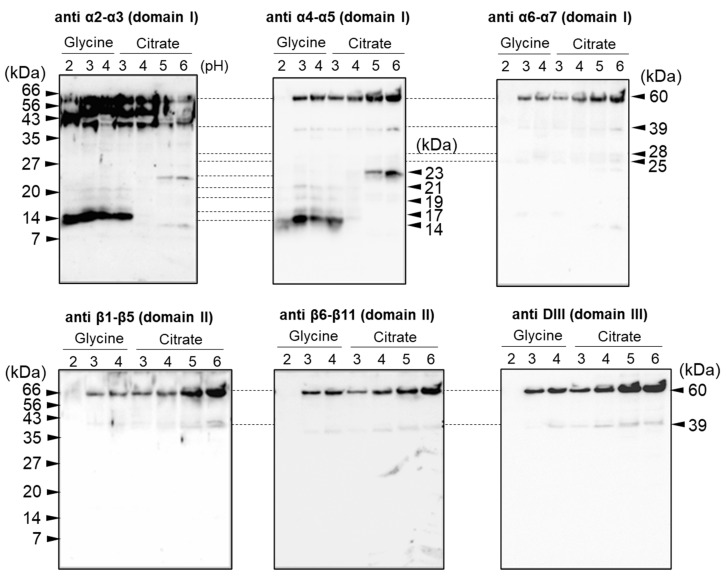
Western blotting analysis of Cry1Aa digests. Cry1Aa digests generated by SGF treatment were separated by 14% SDS-PAGE and then electroblotted onto PVDF membranes. Cry1Aa digests on the membrane were analyzed by Western blotting with six different antisera specific for α2–3, α4–5, and α6–7 helices of domain I, β1-5 and β6-11 sheets of domain II, and domain III.

**Figure 3 biology-14-00015-f003:**
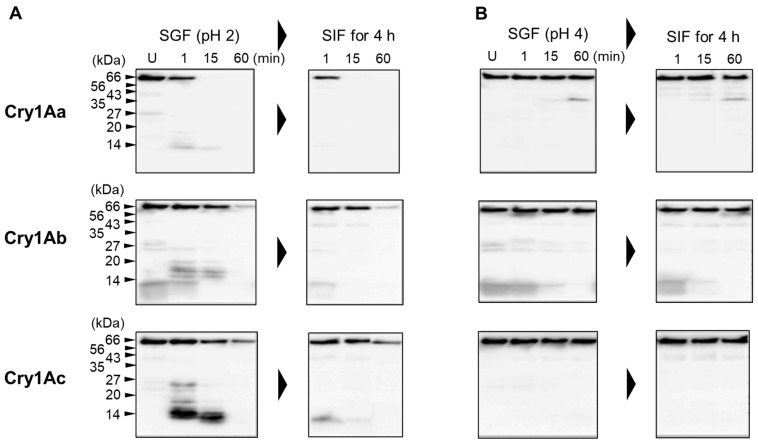
Western blotting analysis of Cry1A digests. Cry1A digests generated by SGF treatment (pH 2 and 4) and digests generated by subsequent SIF treatment (pH 8) were separated by 14% SDS-PAGE and then electroblotted onto PVDF membranes. Cry1A digests were analyzed by Western blotting using anti-α4–α5 antiserum. U: undigested Cry1A toxin. (**A**) SGF digestion at pH 2. (**B**) SGF digestion at pH 4.

**Figure 4 biology-14-00015-f004:**
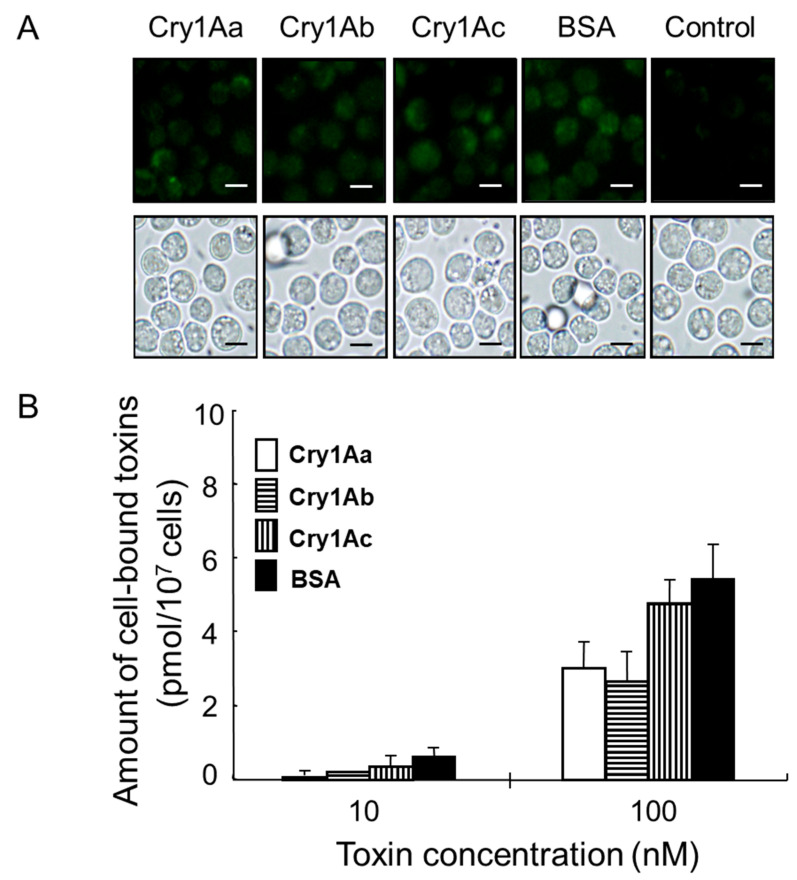
Treatment of rat mast cells with Cry1A toxins. RBL-2H3 rat mast cells were incubated with labeled Cry1A toxins at 10 and 100 nM for 60 min. BSA was used as a control. (**A**) Microscopic observation of RBL-2H3 rat mast cells. Top: Fluorescence of labeled proteins on cells after treatment with toxin at 100 nM. Bottom: Observation of cells under visual light. Bar, 20 μm. (**B**) Amount of each cell-bound Cry1A toxin, as measured using a fluorescence spectrometer. Standard deviation was calculated from triplicate experiments.

**Figure 5 biology-14-00015-f005:**
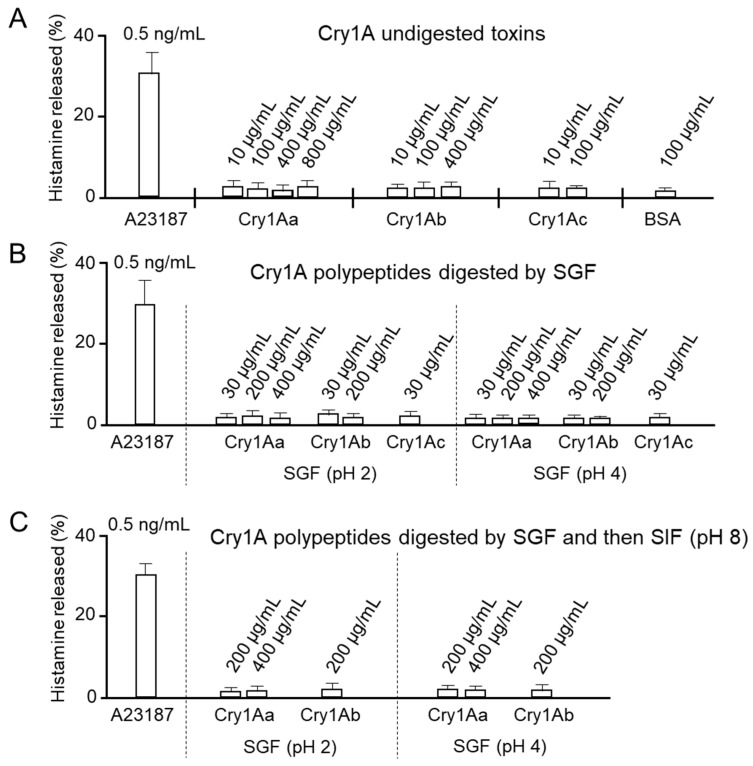
Histamine release from rat mast cells following treatment with Cry1A toxins. (**A**) RBL-2H3 rat mast cells were incubated with undigested Cry1A toxin, and histamine released was quantified using the OPA method. The calcium ionophore A23187 and BSA were used as positive and negative controls, respectively. (**B**) Stimulation of histamine release by Cry1A polypeptides digested with SGF. (**C**) Stimulation of histamine release by Cry1A polypeptides digested sequentially with SGF and SIF.

**Figure 6 biology-14-00015-f006:**
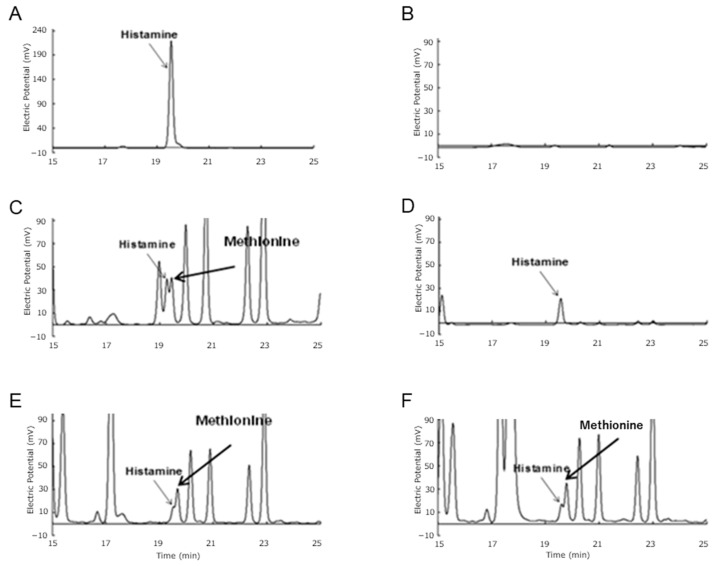
HPLC analysis of histamine released from rat mast cells treated with Cry1Aa digests. Histamine released from rat mast cells was analyzed by HPLC. (**A**) Authentic histamine. A single peak was detected at an RT of approximately 19.6 min. (**B**) Hanks’s buffer. No clearly discernible peak was detected. (**C**) RBL-2H3 rat mast cells. Cells were sonicated in Hanks’s buffer. (**D**) Histamine released from RBL-2H3 rat mast cells treated with ionophore A23187. (**E**) Histamine released from untreated RBL-2H3 rat mast cells. (**F**) Histamine released from RBL-2H3 rat mast cells treated with Cry1Aa digests obtained from SGF treatment.

## Data Availability

Data are available from the authors upon reasonable request.

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
