# Peer review of "Bacillus thuringiensis* Cry1A Insecticidal Toxins and Their Digests Do Not Stimulate Histamine Release from Cultured Rat Mast Cells"

_biology, 2024, doi:10.3390/biology14010015_

Round 1
Reviewer 1 Report
Comments and Suggestions for Authors
1. While the abstract mentions histamine release, it does not specify the key techniques used to measure this (e.g., fluorescence microscopy, HPLC, enzyme-linked assays). Also results should be more technical like Histamine release was significantly higher compared to control? A brief mention of the methods and results can make abstract more technical and concise.
2. In introduction portion authors should add previous research on Cry1A toxins and mast cell biology. For instance, you could mention previous studies that have explored how Cry1A toxins affect other immune cells or their potential roles in allergic reactions.
3. You briefly mention mast cells as a model for studying histamine release. It would be helpful to expand on why mast cells were chosen, such as their known involvement in allergic reactions and their sensitivity to toxins. A sentence like, "Mast cells are a critical component of the immune system, known for their role in allergic reactions and the release of histamine in response to environmental triggers, making them an ideal model for this study" would strengthen this section.
4. In methodology section authors mention the use of an inverted fluorescence microscope to observe cells treated with Cry1A toxins. It would be helpful to expand on the specific settings or conditions of the microscope used, such as magnification settings or camera type, especially for images that show fluorescence data.
5. In statistical section authors mentions that standard deviations were calculated for fluorescence intensity measurements (e.g., for the microscopy section). However, did not mention how the data were analyzed statistically (e.g., using t-tests, ANOVA).
6. In Figure 4 A add the scale bar
7. In conclusion simplify sentences to make the conclusion more concise without losing depth also clarify the findings by adding significant observations.
8. Add SDS-PAGE and western blotting original figures without processing in supplementary file.
Author Response
- While the abstract mentions histamine release, it does not specify the key techniques used to measure this (e.g., fluorescence microscopy, HPLC, enzyme-linked assays). Also results should be more technical like Histamine release was significantly higher compared to control? A brief mention of the methods and results can make abstract more technical and concise.
As suggested by the reviewer, we have modified the abstract. See page 1, lines 38-43.
- In introduction portion authors should add previous research on Cry1A toxins and mast cell biology. For instance, you could mention previous studies that have explored how Cry1A toxins affect other immune cells or their potential roles in allergic reactions.
As suggested by the reviewer, we have added information on the interaction between Cry1A toxin and the allergy-related immune response. See page 2, lines 100-102.
- You briefly mention mast cells as a model for studying histamine release. It would be helpful to expand on why mast cells were chosen, such as their known involvement in allergic reactions and their sensitivity to toxins. A sentence like, "Mast cells are a critical component of the immune system, known for their role in allergic reactions and the release of histamine in response to environmental triggers, making them an ideal model for this study" would strengthen this section.
Thank you very much for your suggestion. We have added the sentence at the end of the introduction section. We feel the sentence has made this section much better. See page 2, lines 111-114.
- In methodology section authors mention the use of an inverted fluorescence microscope to observe cells treated with Cry1A toxins. It would be helpful to expand on the specific settings or conditions of the microscope used, such as magnification settings or camera type, especially for images that show fluorescence data.
In light of the reviewer’s comment, we have added additional explanation. See page 4, lines 174-175.
- In statistical section authors mentions that standard deviations were calculated for fluorescence intensity measurements (e.g., for the microscopy section). However, did not mention how the data were analyzed statistically (e.g., using t-tests, ANOVA).
In light of the reviewer’s comment, we have added a sentence in the M&M section. See page 4, lines 177-178 and 197.
- In Figure 4 A add the scale bar
In light of the reviewer’s comment, we have added a scale bar in Figure 4A.
- In conclusion simplify sentences to make the conclusion more concise without losing depth also clarify the findings by adding significant observations.
As suggested by the review, we have reorganized sentences in the conclusion section. See page 11, lines 563-572.
- Add SDS-PAGE and western blotting original figures without processing in supplementary file.
Actually, we did not process the original figures except for the color mode of SDS-PAGE. Just in case, we replaced SDS-PAGE figure with gray scale to that of the original color. We also put original figures into supplementary file.
Reviewer 2 Report
Comments and Suggestions for Authors
The authors provided a manuscript dedicated to the study of allergenecity of Cry1A toxins from Bacillus thuringiensis. The toxins are widely used as natural pesticide therefore the study of the impact on mammalian organisms is very important. Some comments regarding the manuscript:
1. The author make accent on the BT-GMO plants, though using bacterial produced proteins in the study. The comparison is not correct. Proteins originated from plants should be used.
2. Two Cry toxins of three are produced by Bt and one is produced by E. coli. The origin should be the same. It is strange to compare non-recombinant and recombinant proteins.
3. SIG treatment should be carried out at higher pH than SGF. How was it verified? What was the final pH in the reaction solution?
4. In the text 3.1 it is mentioned that citrate was used for pH 2-4 and glycine pH 3-6, though it is the vice versa at the figure. Please, correct.
5. Usual SGF digestion time is 1h, why 15 min was used for 3.1?
6. No control is provided for SIF treatment. Does it really work?
7. In 3.4 section untreated toxins are used, though the results of SGF and SIF digestions should be also used.
8. At fig 5. why different sets of protein concentrations are used? And why no Cry1Ac digested with SGF+SIF are tested?
9. The discussion section is mostly the repetion of the introduction and should be rewritten.
Author Response
- The author make accent on the BT-GMO plants, though using bacterial produced proteins in the study. The comparison is not correct. Proteins originated from plants should be used.
As you may know, field cultivation of GMOs is not allowed in Japan, so it is difficult to obtain BT-GMO. Therefore, in this study, we would like to use toxins produced by bacteria instead of those from BT-GMO. We believe that toxins expressed in BT-GMO plants can be replaced by toxins produced by bacteria.
- Two Cry toxins of three are produced by Bt and one is produced by E. coli. The origin should be the same. It is strange to compare non-recombinant and recombinant proteins.
In general, many Bt strains produce multiple toxins. To avoid cross-contamination of toxins, Bt strains that produce only one type of toxin were used in this study. On the other hand, we do not have a Bt strain that produces only Cry1Ab; a recombinant E. coli was used to produce it. See page 3, lines 118-124.
- SIG treatment should be carried out at higher pH than SGF. How was it verified? What was the final pH in the reaction solution?
In light of the reviewer’s comment, we have clarified sentences in section 2.2. See page 3, lines 149-150.
- In the text 3.1 it is mentioned that citrate was used for pH 2-4 and glycine pH 3-6, though it is the vice versa at the figure. Please, correct.
Thank you very much. We have fixed as the reviewers commented. See page 5, line 212.
- Usual SGF digestion time is 1h, why 15 min was used for 3.1?
We understand your concern. If we were only monitoring the sensitivity of each Cry1A toxin to SGF digestion, a 1h SGF digestion might be more appropriate than a 15 min digestion. However, in this study we wanted to detect a variety of potential undigested Cry1A polypeptides and include them in the allergenicity test using mast cells. Therefore, we used 15 min as the digestion time.
- No control is provided for SIF treatment. Does it really work?
We use BSA as a control for the digestion experiments. Because BSA is highly sensitive to SGF under physiological conditions, we were unable to prepare a sufficient amount of protein sample for subsequent SIF treatment. The experiment using SIF was to observe further digestion of the polypeptides generated by SGF treatment, and we believe that a control such as BSA is not necessary.
- In 3.4 section untreated toxins are used, though the results of SGF and SIF digestions should be also used.
As mentioned by the reviewer, we used only untreated intact toxin for fluorescence microscopy observation, not digests by SGF and SIF treatment. This was because we were unable to prepare sufficient amounts of Atto-645-labeled SGF and SIF digests. We believe that the labeling efficiency of the fluorescent dye Atto-645 was affected by protease in SGF and SIF samples. Since no specific binding of intact toxins to cultured mast cells was observed, we believe that similar results will be observed for digests. Indeed, no significant histamine release was detected from mast cells treated with intact toxins and these digests.
- At fig 5. why different sets of protein concentrations are used? And why no Cry1Ac digested with SGF+SIF are tested?
As mentioned on page 8, lines 415-418. We often observed significant aggregation, especially at high concentrations of Cry1Ab and Cry1Ac. We avoid the experiment if aggregation was observed. We have clarified the sentence. See page 8, line 411.
- The discussion section is mostly the repetion of the introduction and should be rewritten.
As suggested by the reviewer, we have reorganized sentences in the discussion and introduction section. See page 2, lines 90-105, and page 10, lines 516-517, 536-539.
Round 2
Reviewer 1 Report
Comments and Suggestions for Authors
No further comments paper can be accepted.
Author Response
We would like to thank the reviewer for many useful comments and suggestions.
Reviewer 2 Report
Comments and Suggestions for Authors
I would like to thank the authors for addressing my comments. Though some of my remarks remain not fully answered.
1. I do understand that there might be some legal difficulty with study GMO plants directly. But bacteria and plants have very different systems of protein synthesis and processing, including post-translational modifications like glycosilation which drastically affect the allergenicity of the proteins. You should either use plant-originated protein or not extrapolate results to the GMO plants in the conclusion.
2. In Bt Cry proteins are produced during sporulation phase and then released in the medium during mother cell lysis, E. coli does not have this phase and tend to accumulate the recombinant proteins in inclusion bodies. So, the processes are quite different. You have provide the protocols of the protein extraction and test whether the protein from E. coli is functional.
3. Ok
4. Ok
5. Sure, but you make a conclusion regarding the allergenicity of the digested proteins. And the the proteolytic fragments might have more allergic potential than the full-length proteins, so different digestion periods should be tested.
6. I would not mention the control if any signs of digestion could be seen at figure 3. But all the proteins seems absolutly unchanged. That's why the control should be done.
7. Proteolytic fragments and the full-length proteins have absolutly different structure, so they might have different binding abilities. So, you cannot make any conclusion regarding the binding of proteins after SGF and SIF.
8. Aggregation properties of proteins and allergenicity of proteins can be related. So, aggregated fraction should also be tested. Moreover, the aggregates can be formed during real digestion in the human intestine.
Author Response
Comment 1. I do understand that there might be some legal difficulty with study GMO plants directly. But bacteria and plants have very different systems of protein synthesis and processing, including post-translational modifications like glycosilation which drastically affect the allergenicity of the proteins. You should either use plant-originated protein or not extrapolate results to the GMO plants in the conclusion.
In light of the reviewer’s comment, we have modified sentences in both “Simple summary” and “Abstract”. See page 1, lines 28-30, and 43-44.
Comment 2. In Bt Cry proteins are produced during sporulation phase and then released in the medium during mother cell lysis, E. coli does not have this phase and tend to accumulate the recombinant proteins in inclusion bodies. So, the processes are quite different. You have provide the protocols of the protein extraction and test whether the protein from E. coli is functional.
Actually, our method is conventional one and the Cry1A toxins prepared using this method are highly active against silkworm (Bombyx mori larvae) and/or diamondback moth (Plutella xylostella).
In light of the reviewer’s comment, we have included an additional explanation in the M&M section. See page 3, lines 125-127.
Comment 3. Ok
Comment 4. Ok
Comment 5. Sure, but you make a conclusion regarding the allergenicity of the digested proteins. And the the proteolytic fragments might have more allergic potential than the full-length proteins, so different digestion periods should be tested.
We chose 15 min incubated Cry1A toxin samples for this study because the difference in SGF sensitivity of Cry1A toxins was clear and the digested polypeptide samples could be recovered efficiently. We believe that SGF treatment for 15 min can produce a variety of polypeptides on the way to complete digestion. In fact, if we look at Figure 2 (section 3.2.), many smaller polypeptides were observed that were not detected by SDS-PAGE (Figure 1).
In light of the reviewer’s comment, we have included an additional explanation in the Results section. See page 5, lines 224-228.
Comment 6. I would not mention the control if any signs of digestion could be seen at figure 3. But all the proteins seems absolutly unchanged. That's why the control should be done.
In Figure 3, it appears that the polypeptides generated by the SGF digestion (especially pH 2 of SGF) are further digested by the SIF treatment. However, intact toxins and some polypeptides remain undigested. As mentioned in the M&M section, Cry1A toxins is generally activated by trypsin treatment. Intact toxin and presumably some digested toxin fragments may be relatively resistant to SIF digestion.
In light of the reviewer’s comment, we have included an additional explanation in Section 3.3. See page 7, lines 315-321.
Comment 7. Proteolytic fragments and the full-length proteins have absolutly different structure, so they might have different binding abilities. So, you cannot make any conclusion regarding the binding of proteins after SGF and SIF.
We agree that the intact toxin and the proteolytic fragments have different structures. Although we could not prepare technically the Atto-645-labeled SGF and SIF digests, it is of interest to investigate the interaction.
In light of the reviewer’s comment, we have included an additional explanation in Section 3.4. See page 8 lines 367-370.
Comment 8. Aggregation properties of proteins and allergenicity of proteins can be related. So, aggregated fraction should also be tested. Moreover, the aggregates can be formed during real digestion in the human intestine.
In this study, significant aggregation of Cry1A and Cry1A digests was observed, and we avoided using such a turbid toxin solution for histamine release analysis. Since Cry1A toxin is expected to be in a soluble form, we found it difficult to evaluate the result using an aggregated sample. On the other hand, as mentioned by the reviewer, we believe that it remains possible that such an aggregated protein sample could induce histamine release from mast cells.
In light of the reviewer’s comment, we have included an additional explanation in the Discussion section. See page 11 lines 565-569.